# Birhan maternal and child health cohort: a study protocol

Grace J Chan [ID],[1,2,3] Bezawit Mesfin Hunegnaw,[2] Kimiko Van Wickle,[3] Yahya Mohammed,[4] Mesfin Hunegnaw,[4] Chalachew Bekele,[4] Frederick G B Goddard,[3] Fisseha Tadesse,[5] Delayehu Bekele[3,6]

[1]Department of Pediatrics, Boston Children's Hospital, Harvard Medical School, Boston, MA, USA
[2]Department of Pediatrics and Child Health, St Paul's Hospital Millennium Medical College, Addis Ababa, Ethiopia
[3]Department of Epidemiology, Harvard University T H Chan School of Public Health, Boston, Massachusetts, USA
[4]HaSET, St Paul's Hospital Millennium Medical College, Addis Ababa, Ethiopia
[5]Department of Obstetrics and Gynecology, Debre Birhan Referral Hospital, Debre Birhan, Ethiopia
[6]Department of Obstetrics and Gynecology, Saint Paul's Hospital Millennium Medical College, Addis Ababa, Ethiopia

**Correspondence to**
Dr Grace J Chan;
grace.chan@hsph.harvard.edu

## ABSTRACT

**Introduction** Reliable estimates on maternal and child morbidity and mortality are essential for health programmes and policies. Data are needed in populations, which have the highest burden of disease but also have the least evidence and research, to design and evaluate health interventions to prevent illnesses and deaths that occur worldwide each year.

**Methods and analysis** The Birhan Maternal and Child Health cohort is an open prospective pregnancy and birth cohort nested within the Birhan Health and Demographic Surveillance System. An estimated 2500 pregnant women are enrolled each year and followed through pregnancy, birth and the postpartum period. Newborns are followed through 2 years of life to assess growth and development. Baseline medical data, signs and symptoms, laboratory test results, anthropometrics and pregnancy and birth outcomes (stillbirth, preterm birth, low birth weight) are collected from both home and health facility visits. We will calculate the period prevalence and incidence of primary morbidity and mortality outcomes.

**Ethics and dissemination** The cohort has received ethical approval. Findings will be disseminated at scientific conferences, peer-reviewed journals and to relevant stakeholders including the Ministry of Health.

## Strengths and limitations of this study

► This cohort collects longitudinal data at multiple time points from pregnancy through birth and childhood in a setting where there are limited data.
► The longitudinal nature of the cohort will support time-to-event analysis.
► The study generates accurate estimates of birth outcomes such as stillbirths, preterm birth and low birth weight with near complete follow-up, gestational age dating with ultrasound and digital weight measures for community, home and facility births.
► Similar to all observational studies, there are potential confounders that are unmeasurable.
► As with any longitudinal study, there is study participant attrition and missing data will be managed with imputation methods during analysis.

## INTRODUCTION

Globally, there has been significant progress in improving maternal and child health over the past few years. However, maternal, neonatal and child mortality remains disproportionately high in parts of the world that have limited data, research and programmes. Almost 94% of maternal mortality occurs from preventable causes in low- middle-income country (LMIC) settings.[1] Sub-Saharan Africa has the highest under-5 mortality.[2] There is a significant gap in our knowledge of stillbirths rates and where still births occur. An estimated 2.6 million (uncertainty range 2.4–3.0 million) third trimester stillbirths occurred in 2015 worldwide with 98% estimated to have occurred in LMIC settings.[3 4]

In Ethiopia, the second most populous country in Africa, the maternal mortality rate is 412/100 000 live births totalling 14 000 maternal deaths each year.[5–7] Ethiopia is one of five countries which contributed to more than half of the global under 5 deaths in 2018.[8] With 80 000 deaths each year, Ethiopia is 1 of 10 countries accounting for more than half of global neonatal deaths.[9] The rate of stillbirths at a population level in Ethiopia remains unmeasured. Rough estimates range from 9.2 stillbirths per 1000 births, based on data from the 2016 Ethiopia Demographic and Health Survey maternal report of a pregnancy loss after 7 months gestation in the last 5 years, to 28 stillbirths per 1000 births from a 2014 systematic review of predominantly hospital-based studies.[10–12]

Priority setting for policies, programmes and research to prevent adverse maternal and neonatal outcomes relies on accurate and timely health data.[13] Data are needed for the implementation of health interventions to prevent maternal and child morbidity and mortality. However, data on basic vital statistics are lacking in most low-income countries. In sub-Saharan Africa, for example, fewer than 10 countries have vital registration systems that produce usable data.[14]

There are significant gaps in counting every pregnancy, newborn and child; identifying high-risk populations; and understanding

causal pathways for morbidity and mortality to develop and test interventions. Many maternal and neonatal deaths are preventable with improved identification of high-risk pregnancies, treatment and management of maternal complications, and improved quality of care at birth and during the postnatal and early childhood period.[3 4] Access to timely, accurate and dependable information is essential to improving health and health systems. To address these gaps, the Birhan Maternal and Child Health (MCH) cohort is designed to describe the epidemiology of maternal and childhood health outcomes and risk factors in North Shewa Zone, Amhara Region, Ethiopia. Specific objectives include:

1. To estimate the incidence of birth outcomes, including stillbirths, early neonatal mortality, preterm birth and low birth weight.
2. To identify severe illnesses and deaths among pregnant women and children under 2 years old and identify risk factors for morbidity and mortality.
3. To build the capacity of the Ethiopian Federal Ministry of Health and local institutions to generate and use local evidence for action to improve maternal and child health.

## METHODS AND ANALYSIS
### Study design
The Birhan MCH cohort is an open prospective pregnancy and birth cohort nested within the Birhan Health and Demographic Surveillance System (HDSS). Briefly, the Birhan HDSS includes 16 kebeles (lowest administrative unit) in two districts in Amhara, Ethiopia. As of July 2019, the HDSS includes a total of 18 933 households and 77 766 people.[15] The site includes eight health facilities (three hospitals and five health centres), where all health services for pregnant women, postpartum women and children under-5 are provided free of charge by the Ministry of Health. The Birhan MCH cohort started in December 2018 and will continue through 2024 with plans to continue as funding allows. Basic health and demographic data are collected every three months through house-to-house surveillance. During these quarterly visits, data collectors conduct pregnancy surveillance among married women of childbearing age through a series of pregnancy screening questions. For women who screen positive, pregnancies are confirmed with urine HCG tests. Non-married women are not screened for pregnancy to respect cultural norms; non-married women who are visibly pregnant may be consented. Pregnant women who consent to participate are enrolled into the MCH cohort. Mothers of and children under -2 are also consented and enrolled into the MCH cohort. Both clinical and epidemiological data are collected at health facilities during antenatal, postnatal and outpatient sick visits and in the community during scheduled home visits.

### Study population
Pregnant women, mothers of children under-2 years of age and children under-2 from the Birhan catchment population are eligible for enrolment if they meet the following inclusion criteria:
- ▶ Provide informed consent.
- ▶ Enrolled in the Birhan HDSS. Pregnant women must have a confirmed pregnancy (urine pregnancy test, ultrasound or other positive signs of pregnancy).

Women and children are excluded from the study if they are from:
- ▶ Semiformed populations living in camps, for example, military personnel and prisoners.
- ▶ Street children or orphans who have no permanent address.
- ▶ Those who come to the area only for work during the day and have a primary household outside the catchment area.

### Follow-up schedule
Once enrolled, pregnant women are followed both at home and at the health facility. Home visits are conducted every three months until 32 weeks of gestation, then every two weeks. After 36 weeks, home visits alternate with phone visits every week until birth. Postpartum women are followed from birth to 42 days post partum with scheduled home visits on days 0 (birth), 6, 28 and 42 after birth. Newborns and children under-2 years are followed on days 0 (birth), 6, 28 and 42 and months 6, 12 and 24. Participants who present to health facilities for antenatal care, postnatal visits or sick visits (outpatient or admission) also have data collected by study data collectors at the health facility. Participants are censored from the study with the following events: out-migration from the catchment area, death, lost to follow-up and at the end of the follow-up period (figure 1). In addition, all participants (including pregnant women, postpartum women and children under-2) are concurrently followed on a quarterly basis by the underlying HDSS, which includes basic health questionnaires and an under-2 year old questionnaire on child morbidity.

### Data collection
Data are collected by several sources: maternal or caretaker recall, data collector assessments in the community, health worker observations at the community, health centre and hospital level. Data collected at the time of HDSS enrolment include household data on socioeconomic status, drinking water and sanitation access, flooring/wall/roof materials, number of rooms used for sleeping, place of cooking, household possessions and ownership of land. Individual (mother, father, child and grandparent) specific data on education, employment, literacy, religion, care seeking behaviours and immunisation rates are collected. Anthropometric data including weight, height and mid-upper arm circumference (MUAC) measurements are collected for women of childbearing age and children under-2 following standardised

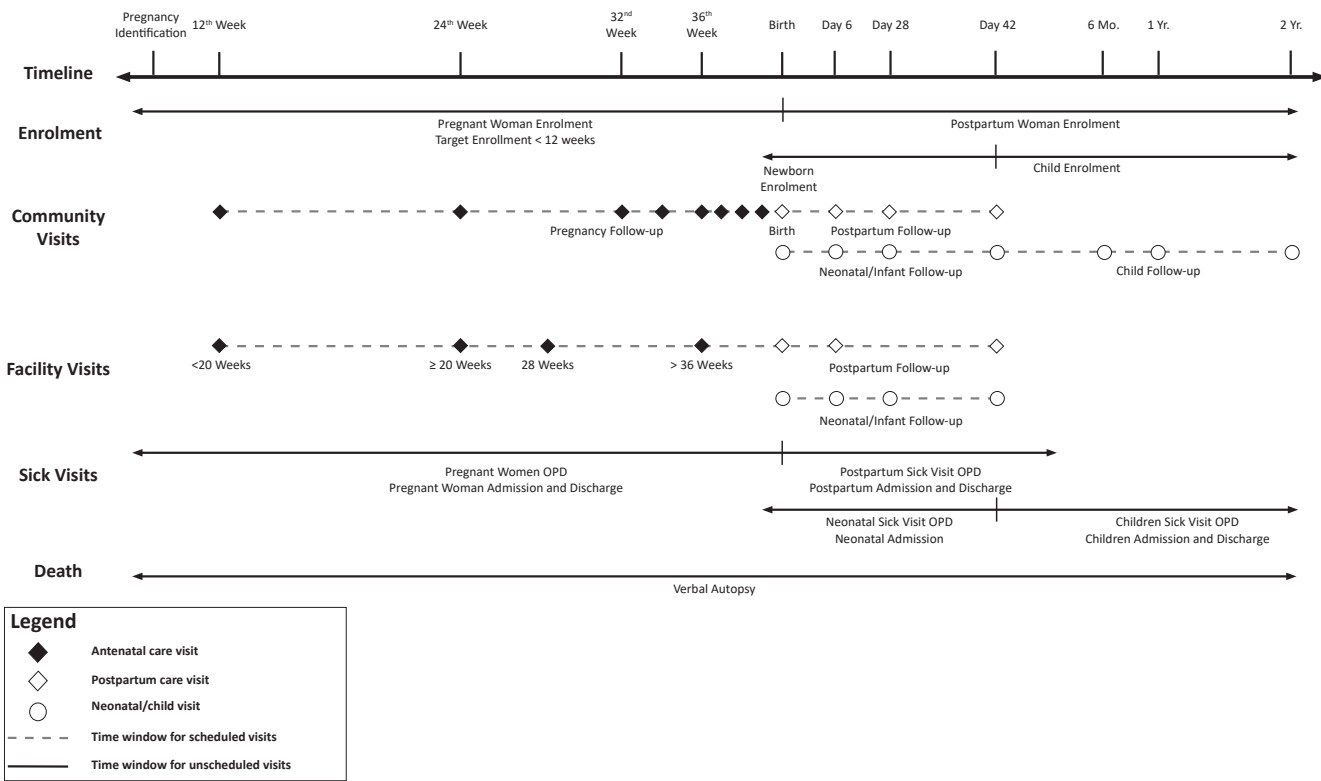

**Figure 1** Schedule of study enrolment and follow-up visits ◆ Antenatal care visit. ◇ Postpartum care visit. ○ Neonatal / child visit. ------ Time window for scheduled visits. ——— Time window for unscheduled visits. OPD; Outpatient Department.

procedures and equipment: Seca 874 digital flat scale, Seca 354 digital baby scale, Seca 417 infantometers and tape measures. Data collectors were recruited from the local villages with at least a diploma in clinical nursing or midwifery. Supervisors have at least a master's in public health. For study purposes, data collectors received four weeks of intensive training with lectures, interactive discussions on the study protocol coupled with practical training sessions on clinical assessments and anthropometric measurements. Following training, there is onsite supervision and mentorship every week.

Pregnant women enrolled into the cohort receive an ultrasound for gestational age dating at their earliest antenatal care (ANC) visit. Information on the pregnancy (birth spacing, antenatal care visits) and pregnancy-related symptoms, including history of vaginal bleeding, dysuria, headache, fever, abdominal pain and fetal movement are collected. Maternal weight and observations of clinical signs such as pallor, jaundice and oedema are collected. For newborns enrolled in the cohort, data on labour and delivery (location of delivery, delivery method, duration of labour, complications of birth, birth weight) and immediate newborn care (breast feeding, cord-care, bathing, skin emollients) are collected as soon as possible with a target of within 24 hours of birth. Mothers or caretakers recall of clinical symptoms, observations of signs of illness and newborn weight are collected at subsequent postpartum visits. Sick visits include outpatient clinic visits

and hospital admissions. Clinical symptoms, observations of signs of illness and possible risk factors such as sick contacts, food sources, travel and water sanitation and hygiene practices are collected at sick visits. Additionally, data on treatment modalities, referrals and vital status are collected at discharge.

To ascertain the cause of death, verbal autopsies (VAs) are conducted to obtain information on the symptoms, signs and other relevant events during the illness leading to death. Three VA questionnaires (for deaths 0–28 completed days of life; deaths of children between four weeks and 11 years of age; and deaths of persons aged 12 years and above which includes pregnancy-related questions for women of reproductive age) adopted from the standard 2016 WHO VA questionnaires are used.[16] Verbal autopsies were developed to provide information on causes of death in communities where there is limited access to healthcare and medical certification of causes of death. In such situations, the main source of information about the event is from caregivers of the deceased, most often family members. Ascertaining causes of death from such information is based on the premise that VA respondents can accurately recall details of the symptoms and events that occurred during the period of illness prior to death, and that such information can be used to classify the cause(s) of death into diagnostic categories based on specific symptoms.

## Electronic data collection system and data management

We use the Birhan electronic data collection system, built from Open Data Kit (ODK),[17] for longitudinal and relational data collection. ODK is a free, open-source application used to facilitate mobile data capture. ODK can be coded using SQL to facilitate data collection, transfer and storage, as well as the development of electronic questionnaires used for data collection. Data can be collected on a mobile device offline. Data are uploaded daily using mobile data to a central database. Data quality checks are built into the data entry tools and data system. Data are hosted on an encrypted central database on the cloud and are deidentified. The data system is developed and maintained by two data system developers. Data are managed by a team of data scientists and data managers in Stata (V.17.0) and R/RStudio (V.4.0.5).[18 19]

## Data quality

To ensure high data quality, we developed simple user-friendly questionnaires which were piloted prior to use. We recruited and trained data collectors who met a minimum level of competency as described above and through pre–post training examinations. Supervisors oversee study implementation by supporting data collectors with weekly on-the site mentorship and weekly team meetings. Supervisors independently conduct home visits on a 5% random sample of households to validate data collector performance. Within the electronic data capture system, we built single field value checks, interfield logic checks and interform logic checks. In case of an error at the time of data entry, pop-up warnings are triggered, prompting the user to resolve any issues prior to saving the record. Following data collection, the dataset is systematically checked for data quality issues by a team of analysts. Data quality issues are recorded, provided to data collectors to identify solutions and then rectified in the dataset.

## Primary outcomes

Primary outcomes include:
- ▶ Severe maternal morbidity and maternal mortality: unintended outcomes of the process of labour and delivery that result in significant short-term or long-term consequences to a woman's health.
- ▶ Stillbirth: death prior to delivery of a fetus ≥28 weeks of gestation (or >1000 g if gestational age is unavailable).
- ▶ 7-day neonatal mortality: death of a live-born infant at 0–6 days of life.
- ▶ 28-day neonatal mortality: death of a live-born infant at 0–27 days of life.
- ▶ Preterm birth: delivery prior to 37 completed weeks of gestation of a live born infant.
- ▶ Low birth weight: birth weight <2500 grams of live-born infant.
- ▶ Place of birth: home or facility.
- ▶ Child morbidity: incidence of diarrhoea, pneumonia, febrile illnesses.[20 21]

- ▶ Causes of mortality: numbers and rates of maternal, childhood and adult deaths and causes of death by VA.[22 23]

Secondary outcomes include recurrence of illnesses; readmissions and growth as measured by weight, height, MUAC and body mass index; maternal and child immunisations, and exclusive breast feeding.

## Data analysis

For each of the primary outcomes, we will calculate the period prevalence and incidence at community and health facilities levels. To estimate period prevalence, we will sum the 2-week periods over all the participants within certain age ranges as the denominator and sum the number of episodes within that 2-week period as the numerator (i.e., cases over sum of person-days observed or person-days recalled). We will estimate the incidence for each outcome by taking the sum of all the person-time contributed by each person as the denominator, and the sum of all episodes over the study period as the numerator. Missing data will be assessed through quality assurance checks by field supervisors and rectified through a documented error correction system. Remaining incomplete data will be addressed through analytical approaches. We will repeat the above analyses with varying case definitions of disease severity.

In a population of 77 000, we expect approximately 2500 pregnancies per year and 4500 children under-2 years of age. Using historical data and the literature, we expect a range of outcome rates from 2.0% to 20.0% depending on outcome of interest (table 1). The prevalence of stillbirths is estimated to be 9.2 per 1000 births in Ethiopia and 19.7 per 1000 births (around 2.0%) in Amhara.[24] The prevalence of diarrhoea over a 2-week period among children under-2 varies by region: 31.3%[25] in Afar, 18.5% in Southern Nations, Nationalities, and People's Region (SNNPR), 16.0% in Oromia, 17.7% in Amhara and 6.8% in Tigray.[26] Given this variation, we estimate the prevalence of diarrhoea over a 2-week period among children under-2 to be 20.0%. We expect the prevalence of acute respiratory infection over a 2-week period among children under-2 to be 5.0% based on preliminary data in Ethiopia.[26]

**Table 1** Outcomes of interest, estimated prevalence and precision (95% CI width) by sample size*

| Outcomes | Estimated prevalence | 2500 | 3500 | 4500 |
|---|---|---|---|---|
| Stillbirth[24] | 2.0% | 0.6% | 0.5% | 0.4% |
| Neonatal death (28 days)[5] | 3.0% | 0.7% | 0.6% | 0.5% |
| Preterm birth | 12.0% | 1.3% | 1.1% | 0.9% |
| Diarrhoea†[26] | 20.0% | 1.6% | 1.3% | 1.2% |

*Precision calculated using exact distribution for <5% prevalence and Wald distribution for ≥5% prevalence.
†Estimated 2-week point prevalence for children under 2 years old.

To estimate absolute precision (defined as the half-width of the 95% CI) for outcomes ranging in prevalence from 2.0% to 20.0%, the exact (Clopper-Pearson) distribution was used for rare outcomes (<5%) and the Wald distribution was used for common outcomes (≥5%).[27 28] A sample size of 2500 pregnant women would estimate the prevalence for a rare outcome such as stillbirth, 2.0%, with 0.6% precision (95% CI 1.4% to 2.6%) (table 1). For more common outcomes such as preterm birth, a sample size of 2500 pregnant women would estimate a 12.0% prevalence with 1.3% precision (95% CI 10.7% to 13.3%). A sample size of 4500 children would estimate a 20.0% prevalence of diarrhoea with 1.2% precision (95% CI 18.8% to 21.2%). A sample size of 4500 children would estimate a 5.0% prevalence of ARI with 0.5% precision (95% CI 4.5% to 5.5%). Period prevalence will vary based on case definitions and severity of illnesses. To examine risk factors and correlates for these outcomes, we will conduct tests of association to detect statistically significant effect sizes for risk factors and outcomes assessed in this study.

## ETHICS AND DISSEMINATION

The study has received the following ethical approvals: St. Paul's Hospital Millennium Medical College (PM23/274), Boston Children's Hospital (P00028224) and Harvard School of Public Health (19-0991). Results will be made available to the Ministry of Health, regional, zonal and district health offices. Birhan serves as the field site for HaSET ('happiness' in Amharic)—a maternal, newborn and child health research programme in Ethiopia. For further details, please visit www.hasetmch.org. Publications will follow HaSET publications guidelines.

### Patient and public involvement

The scope of the research questions and outcome measurements were informed by identified gaps in maternal and child health and key priorities of the Ministry of Health, including limited reliable health and demographic data and data on health outcomes, which are needed to support evidence-based decision making. To build capacity on research methodologies, data analyses and interpretation of data, fellows and trainees are involved in the cohort study. A community advisory board is engaged with the research.

### Access to data

Data use is governed by the Birhan Data Access Committee (DAC) and follows Birhan's data sharing policy. All researchers who wish to access Birhan data can complete a Birhan data request form and submit it for decision by the Birhan DAC. Datasets will only be provided with deidentified data to maintain confidentiality of study participants.

**Acknowledgements** We are grateful to the community of the Birhan field site, especially the mothers and children who participate. We appreciate the generous support from the Ministry of Health, St. Paul's Hospital Millennium Medical College,
Amhara Regional Health Bureau, North Shewa Zone Office, Angolela Tera and Kewet/Shewa Robit Woreda Health Offices and catchment health facilities. We thank the data collectors, supervisors, coordinators and the HaSET team for their significant contributions to the Birhan MCH study.

**Contributors** The study concept was conceived by GJC and DB. GJC, DB, BMH and FT contributed with study design and implementation. MH and CB led the data collection. YM, KVW and FGBG contributed to data management. Analysis will be performed by YM, KVW and FGBG. GJC drafted the initial manuscript. All authors contributed to the refinement of the protocol, provided edits and critiqued the manuscript for intellectual content.

**Funding** This work has been supported by the Bill & Melinda Gates Foundation (grant number INV-010382 and OPP1201842).

**Competing interests** None declared.

**Patient and public involvement** Patients and/or the public were involved in the design, or conduct, or reporting, or dissemination plans of this research. Refer to the Methods section for further details.

**Patient consent for publication** Not required.

**Provenance and peer review** Not commissioned; externally peer reviewed.

**ORCID iD**
Grace J Chan http://orcid.org/0000-0002-2716-1643

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
