## [Reviewer comments · BMJ Open]

ARTICLE DETAILS

TITLE (PROVISIONAL)	Birhan Maternal and Child Health cohort: a study protocol
AUTHORS	Chan, Grace; Hunegnaw, Bezawit; Van Wickle, Kimiko; Mohammed, Yahya; Hunegnaw, Mesfin; Bekele, Chalachew; Goddard, Frederick; Tadesse, Fisseha; Bekele, Delayehu

VERSION 1 – REVIEW

REVIEWER	Alves, João Instituto de Medicina Integral Prof. Fernando Figueira (IMIP), Department of Paediatrics
REVIEW RETURNED	06-Apr-2021

GENERAL COMMENTS	Dear Editor, Thanks for the opportunity to review the manuscript titled, "Birhan Maternal and Child Health cohort: a study protocol" by Chan et al. The manuscript addresses a very important issue, especially for poor and developing countries. The objectives are stated clearly and I have only few comments. Study setting and period: please provide more details. Two districts (?) Hospital, Health Center in the study region (?) Study population: This study will be a prospective pregnancy and birth cohort. Why children under-two will be eligible? A child aged 1 year and 6 months will have a follow-up of only 6 months, unlike others who will be followed for 24 months. Will retrospective data be considered in these children? How will the analysis be done? Furthermore, these children will not estimate the incidence of birth outcomes, stillbirths, early neonatal mortality, preterm birth, low birth weight and severe illnesses and deaths among pregnant women. Follow-up Schedule: Basic health is collected every three months through house-to-house surveillance. Why children under two years will be only followed on months 6, 12, and 24? Only one visit during the second year of life... A flow chart is missing. Data collection: Data will be collected by several sources. Will be any prior training of the all team? How anthropometric data will be collected? How high-risk pregnant women will be followed? How is the prenatal routine? Gestational diabetes screening? Birth weight: "For newborns enrolled in the cohort within 24 hours of birth, data on labor and delivery ... birth weight ... are collected. Mothers or caretakers recall of clinical symptoms, observations of
---

	signs of illness, and neonatal weight are collected at subsequent postpartum visits.” This second birth weight does not seem clear. Sick visits: It deserves more details. How verbal autopsies in pregnant women will be performed? Does it included in deaths of persons aged 12 years and above? Does it need a special model for pregnant women? Secondary Outcomes: I suggest to add immunization (mother and child) and exclusive breastfeeding duration.
--	---

REVIEWER	Fernández-Sáez, José University of Alicante, Public Health
REVIEW RETURNED	20-Apr-2021

GENERAL COMMENTS	Dear authors: I enjoyed reading your paper since I consider Maternity Mortality as an interesting and pertinent topic. It is a public health concern that could be avoidable and requires further research. One problem of the epidemiological studies is the lack of systematic information on the social, political and economic context determinants, which are related to health problems (Walters, V. The Social Context of Women's Health. BMC Women's Health 4, S2 (2004). https://doi.org/10.1186/1472-6874-4-S1-S2). For this reason, I see the publication of this paper necessary. I only have a few minor comments: Introduction If possible, references 3, 4 and 5 could be updated. Paragraph 2: When possible, more current data could be provided. Methods and analysis The text says that quarterly visits are made. Have losses been taken into account? Was a sample size study done? Why have only married women been taken? Yours sincerety
---

VERSION 1 – AUTHOR RESPONSE

Reviewer: 1

Dr. João Alves, Instituto de Medicina Integral Prof. Fernando Figueira (IMIP) Comments to the Author:
Dear Editor,

Thanks for the opportunity to review the manuscript titled, "Birhan Maternal and Child Health cohort: a study protocol" by Chan et al. The manuscript addresses a very important issue, especially for poor and developing countries. The objectives are stated clearly and I have only few comments.

1. Study setting and period: please provide more details. Two districts (?) Hospital, Health Center in the study region (?)

Response: Thank you, in the 'Methods' section we've added the following. "Briefly, the Birhan HDSS includes 16 kebeles (lowest administrative unit) in two districts in Amhara, Ethiopia. As of July 2019, the HDSS includes a total of 18,933 households and 77,766 people.¹⁴ The site includes eight health facilities (three hospitals and five health centers), where all health services for pregnant women, postpartum women, and children under five are provided free of charge by the Ministry of Health. The Birhan MCH cohort started in December 2017 and will continue through 2024 with plans to continue as funding allows."

2. Study population: This study will be a prospective pregnancy and birth cohort. Why children under-two will be eligible? A child aged 1 year and 6 months will have a follow-up of only 6 months, unlike others who will be followed for 24 months. Will retrospective data be considered in these children? How will the analysis be done? Furthermore, these children will not estimate the incidence of birth outcomes, stillbirths, early neonatal mortality, preterm birth, low birth weight and severe illnesses and deaths among pregnant women.

Response: Thank you, children <2 are eligible to estimate child morbidity and mortality at cross-sectional time points. For children, who are left censored, they will be included only in the analysis in which they can contribute person-time.

3. Follow-up Schedule: Basic health is collected every three months through house-to-house surveillance. Why children under two years will be only followed on months 6, 12, and 24? Only one visit during the second year of life...

Response. Thank you for bringing this up. To be efficient as part of the underlying house-to-house surveillance, children <2 year old are followed with a questionnaire on morbidity in addition to the standing 6 month, 12, and 24 visits. If there was report of an illness or event, the MCH data collector would visit the child for a sick visit. To better clarify in the 'follow-up schedule' section, we've added "In addition, all participants (including pregnant women, postpartum women, and children <2) are concurrently followed on a quarterly basis by the underlying HDSS, which includes basic health questionnaires and an under <2 year old questionnaire on child morbidity."

4. A flow chart is missing.

Response: Thank you, if this is referring to Figure 1, it can be found at the end of the manuscript.

5. Data collection: Data will be collected by several sources. Will be any prior training of the all team? How anthropometric data will be collected? How high-risk pregnant women will be followed? How is the prenatal routine? Gestational diabetes screening?

Response: In the 'Data collection' section, we have added further detail, "Anthropometric data including weight, height, and mid-upper arm circumference (MUAC) measurements are collected for women of child-bearing age and children under-two following standardized procedures and equipment: Seca 874 digital flat scale, Seca 354 digital baby scale; Seca 417 infantometers, and tape measures. Data collectors were recruited from the local villages with at least a diploma in clinical nursing or midwifery. Supervisors have at least a master's in public health. For study purposes, data collectors received four weeks of intensive training with lectures, interactive discussions on the study protocol coupled with practical training sessions on clinical assessments and anthropometric measurements. This was followed with onsite supervision and mentorship every week."

We conducted passive facility level follow-up visits, aligned with the standard of care prenatal routine. If women returned to facility for high risk pregnancies, these visits were captured by our data collectors. Gestational diabetes screening is not done per standard of care.

6. Birth weight: "For newborns enrolled in the cohort within 24 hours of birth, data on labor and delivery ... birth weight ... are collected. Mothers or caretakers recall of clinical symptoms, observations of signs of illness, and neonatal weight are collected at subsequent postpartum visits." This second birth weight does not seem clear.

Response: Thank you, birth weight will be collected within 24 hours of birth. There is no second birth weight. We have rewritten this sentence for clarity. "For newborns enrolled in the cohort, data on labor and delivery (location of delivery, delivery method, duration of labor, complications of birth, birth weight) and immediate newborn care (breastfeeding, cord-care, bathing, skin emollients) are collected as soon as possible with a target of within 24 hours of birth for those born in at home."

7. Sick visits: It deserves more details.

Response: Thank you, we added additional details under 'Data Collection', "Sick visits include data on outpatient clinic visits and hospital admissions. Clinical symptoms, observations of signs of illness and possible risk factors such as sick contacts, food sources, travel, and water sanitation and hygiene practices are collected at sick visits. Additionally, data on treatment modalities, referrals and vital status are collected at discharge."

8. How verbal autopsies in pregnant women will be performed? Does it included in deaths of persons aged 12 years and above? Does it need a special model for pregnant women?

Response: We are using the World Health Organization VA forms, including the adult VA form which includes deaths of persons aged 12 years and above. The adult form includes a section on pregnancy related questions for women of reproductive age to identify pregnancy related causes of death. In the data collection section, we clarified that the adult form “includes pregnancy related questions for women of reproductive age”

9. Secondary Outcomes: I suggest to add immunization (mother and child) and exclusive breastfeeding duration.

Response: Thank you for this helpful suggestion. We’ve added these in the section on ‘Secondary Outcomes’.

Reviewer: 2

Dr. José Fernández-Sáez, University of Alicante Comments to the Author:

Dear authors:

I enjoyed reading your paper since I consider Maternity Mortality as an interesting and pertinent topic. It is a public health concern that could be avoidable and requires further research.

One problem of the epidemiological studies is the lack of systematic information on the social, political and economic context determinants, which are related to health problems (Walters, V. The Social Context of Women's Health. BMC Women's Health 4, S2 (2004). <https://doi.org/10.1186/1472-6874-4-S1-S2>).

For this reason, I see the publication of this paper necessary.

I only have a few minor comments:

Introduction

1. If possible, references 3, 4 and 5 could be updated.

Response: Thank you, we’ve updated reference 5 from the preliminary report with the 2016 Ethiopia Demographic and Health Survey. We searched the literature but did not find more recent data on the global prevalence of stillbirths.

2. Paragraph 2: When possible, more current data could be provided.

Response: We searched the literature for more current data with no success. The national estimate of maternal mortality is the most recent as this was unfortunately not updated in the mini DHS 2019.

3. The text says that quarterly visits are made. Have losses been taken into account?

Response: We expect lost to follow-up due to migration as well as missing data if women were not present during the visit (data collectors attempt visits three times during each round). Missing data will be taken into account in the analysis through imputation methods.

4. Was a sample size study done?

Response: Yes, sample size calculations are included in the ‘Analysis’ section.

5. Why have only married women been taken?

Response: In the conservative and predominantly rural society we are working in, it was deemed culturally unacceptable to ask pregnancy related questions and performing pregnancy tests to unmarried women. We do include unmarried women who appear to be visibly pregnant. In the ‘Study design’ section, we have added, “Nonmarried women are not screened for pregnancy to respect cultural practices. Nonmarried women who are visibly pregnant are approached for consent”

VERSION 2 – REVIEW

REVIEWER	Alves, João Instituto de Medicina Integral Prof. Fernando Figueira (IMIP), Department of Paediatrics
REVIEW RETURNED	11-Aug-2021
GENERAL COMMENTS	All questions were answered. Good luck!

REVIEWER	Fernández-Sáez, José University of Alicante, Public Health
REVIEW RETURNED	27-Jul-2021
GENERAL COMMENTS	The authors have answered all my questions and comments